# Assessment of Retinal Vessel Tortuosity Index in Patients with Fabry Disease Using Optical Coherence Tomography Angiography (OCTA)

**DOI:** 10.3390/diagnostics13152496

**Published:** 2023-07-27

**Authors:** Kevin Hangartner, Anahita Bajka, Maximilian R. J. Wiest, Sophia Sidhu, Mario D. Toro, Peter M. Maloca, Sandrine A. Zweifel

**Affiliations:** 1Faculty of Human Medicine, University of Zurich, 8032 Zurich, Switzerland; 2Department of Ophthalmology, University Hospital of Zurich, University of Zurich, 8091 Zurich, Switzerland; 3Faculty of Medicine, University of California San Diego, 5998 Alcala Park, San Diego, CA 92110, USA; 4Chair and Department of General and Pediatric Ophthalmology, Medical University of Lublin, 20079 Lublin, Poland; 5Eye Clinic, Department of Public Health, University Federico II, 80131 Naples, Italy; 6Institute of Molecular and Clinical Ophthalmology Basel (IOB), 4031 Basel, Switzerland; 7Department of Ophthalmology, University Hospital Basel, 4031 Basel, Switzerland; 8Moorfields Eye Hospital NHS Foundation Trust, London EC1V 2PD, UK

**Keywords:** retinal vessel tortuosity, vessel tortuosity index, retinal imaging, optical coherence tomography angiography, Fabry disease, lysosomal storage disorder, alpha galactosidase deficiency

## Abstract

Vessel tortuosity (VT) is a parameter used to assess retinal involvement in patients affected by systemic diseases such as Fabry disease (FD). In this study, we assessed a retinal VT index (VTI) using optical coherence tomography angiography (OCTA) in a group of patients with FD (FD cohort) compared to a healthy control group (HC cohort). This is a single-center, retrospective study analysis of all consecutive patients with genetically tested and confirmed FD who underwent regular ophthalmological visits from December 2017 to January 2020 at the Department of Ophthalmology at the University Hospital of Zurich, Switzerland. VTI was calculated for each OCTA image and the results were compared between FD and HC cohort. A total of 56 participants, 32 (male:female ratio 12:20) in the FD cohort and 24 (male:female ratio 13:11) in the HC cohort. Classic onset was determined in 18 patients. Overall, mean VTI (±SD) was 0.21 (±0.07). Male patients with classic-onset FD had a significantly higher mean VTI (0.33, SD ± 0.35) compared to all other subgroups (*p*-value < 0.05). Further investigations of retinal VTI in patients with FD could be helpful to use OCTA as a noninvasive screening and follow-up modality to assess disease progression in affected patients.

## 1. Introduction

Systemic diseases, including rare genetic diseases, can lead to pathological changes in the retinal vasculature, resulting, for example, in dilated retinal vessels, microaneurysms, neovascularizations, and an increase in vessel tortuosity (VT) [1,2,3]. 

Fabry disease (FD) is a rare inherited, X-linked lysosomal storage disorder with an estimated prevalence of between 1:17,000 and 1:117,000 [4]. In patients with FD, the gene coding the enzyme α-galactosidase (GLA) is altered, resulting in its deficiency or inactivity [1,5,6,7]. The deficient or absent activity of GLA causes an accumulation of globotriaosylceramide (Gb3) in several tissues’ lysosomes and fluids, preferably in vessels and smooth muscle cells due to increased endothelial proliferation. The increased endothelial proliferation leads to alterations in the vessels’ walls and thereby changes the vessels’ structure, including retinal vessels [4]. The accumulation ultimately leads to progressive organ dysfunctions, such as cardiac myopathy, renal failure, and cerebrovascular diseases [8,9]. Regarding the eyes, patients with FD can clinically present with, e.g., cornea verticillata, subcapsular cataract, or retinal VT [9,10]. It has been suggested that VT can be of diagnostic and prognostic value in patients with FD, as VT can be assessed with cost-efficient and noninvasive methods and could help to detect changes in the progression of FD by assessing the dynamic of VT [5,11,12,13,14,15,16]. 

In general, the disease is divided into a classic and late-onset phenotype [8,16]. Patients with the classic phenotype have no detectable activity of GLA, leading to an early and more severe manifestation of the disease, starting in early childhood [9]. Its prevalence is estimated to be between 1:22,000 and 1:40,000 in males [4]. In the late-onset phenotype, residual activity of GLA between 2 and 20% can be detected [17,18]. Thereby, symptoms are often less severe and manifest later in adulthood, around the age of 30 years [19]. Clinically, patients with FD can present with hypohidrosis, painful acroparesthesias, abdominal cramping, and diarrhea, as well as petechiae on the skin and further organ-specific symptoms [4]. Due to the X-linked inheritance, FD is more common in males [9]. Females can be affected as well; however, due to skewed X-chromosomal inactivation, females show a broader range of disease severity [10]. To date, patients with FD cannot be cured. However, enzyme replacement therapy (ERT), applied as an infusion every 2 weeks, offers a treatment option, lowering the disease burden as well as protecting organs from accumulating Gb3 [4,17,20]. Therefore, assessing the dynamic of VT in patients with FD being treated with ERT could help to analyze the therapeutic response of the ongoing treatment.

So far, numerous direct measurements of systemic vascular function have been established [21,22,23]. However, direct measurements have certain limitations, only providing information on a few parameters while being often invasive, costly, highly variable, and requiring elaborate training and experience. Therefore, they are often not useful as a screening modality.

Epidemiological studies have shown that using the comparison of arterial and venous perfusion, retinal vessel calibers, and retinal arteriolar narrowing, retinal vasculature showed alterations dependent on gender, ethnicity, and age [24,25,26,27]. Further studies have reported correlations between systemic vascular diseases compared to changes in the retinal vasculature, namely decreased vessel diameter and VT [2,14,28,29]. Retinal VT is a parameter used to assess the retinal involvement in patients affected by systemic diseases [30]. It has been postulated that venous congestion, retinal ischemia, increased blood flow, rare angiogenesis, and weakening of the vessel walls can result in VT [31,32,33]. 

Retinal imaging allows an accessible way to analyze the vascular network, allowing it to be a time and cost-efficient examination modality [34,35]. The eye itself, with its fragile microvasculature, has been shown to be affected by systemic vascular dysregulation, hence reflecting vascular damage of systemic origin [36,37,38]. Thereby, it stands to reason that changes in retinal vasculature, as mentioned above, can be used as an imaging parameter to detect and observe systemic vascular diseases. 

Over the past decades, several approaches have been assessed to quantify retinal VT using optical coherence tomography angiography (OCTA) with different mathematical models [39,40]. To our knowledge, there is no standardized assessment of retinal VT that ophthalmologists can rely on in daily clinical practice [1,40]. Additionally, there are few objective tools to measure VT, leading to high interobserver variability and poor reproducibility [1,29,41]. An algorithm created by Khansari et al. was used to calculate a VT index (VTI) for each defined vessel in a binarized OCTA image and thereby assess OCTA images quantitatively [42]. VTI is defined as follows: VTI = 0.1 × (M·N·SD_θ_·(L_A_/L_C_)). M stands for the average ratio of centerline length to chord length between two inflection points. L_A_ and L_C_ are the length of the centerline (A) and chord (C). N is the number of critical points where the first derivative of the centerline vanishes. SD_θ_ equates the standard deviation of angle differences between lines tangent to each centerline pixel [42]. 

In this study, we aimed to assess retinal VT in patients with FD (FD cohort) and to compare the findings with a healthy control group (HC cohort), by analyzing a novel VTI using binarized OCTA images of a swept-source device using en face images of the superficial capillary plexus (SCP).

## 2. Materials and Methods

### 2.1. Study Design and Participants

This is a single-center, retrospective analysis of all consecutive patients with genetically tested and confirmed FD who underwent regular ophthalmological visits from December 2017 to January 2020 at the Department of Ophthalmology at the University Hospital of Zurich, Switzerland. Written informed consent was obtained before patients’ and probands’ participation in this study. The institutional review board approved the study (Ethics Committee of the Canton of Zurich, BASEC-Nr.2019-02043) and adhered to the tenets of the Declaration of Helsinki [43].

Patients with genetically confirmed FD, which were 16 years old or older, were included in our FD cohort. FD patients with ocular diseases unrelated to FD, including significant lens opacification, current or previous macular and retinal vascular diseases, diagnosis of glaucoma, congenital eye diseases, and high myopia (>4 diopters) were excluded. Non-age-matched healthy subjects, without systemic vascular or ocular diseases, having a best-corrected visual acuity (BCVA) of logMAR 0.1 or better, were included in the HC cohort.

All study subjects underwent a complete ophthalmological examination, including BCVA, IOP measurement, biomicroscopy of the anterior segment, and funduscopy. BCVA was assessed using glasses with refractory values obtained using an autorefractor (NT-530/510^®^, Nidek Inc., San Jose, CA, USA). IOP measurements were acquired using the same device by Nidek.

Subgroup analysis was performed with a separation between classic and late onset of the diseases as well as between males and females, as the disease is known to be more severe in the classic onset and male patients.

### 2.2. Data Acquisition

Imaging of the retina was performed using spectral domain optical coherence tomography (SD-OCT, Heidelberg Spectralis System, software version 1.9.10.0, Heidelberg Engineering, Heidelberg, Germany) and swept-source OCTA (PlexElite 9000, software version 2.0.1.47652, Carl Zeiss Meditec AG, Munich, Germany).

OCTA volumes of 6 mm × 6 mm × 2 mm centered on the fovea were acquired by a well-trained ophthalmologist (M.W.). Scans with a signal strength of 8 out of 10 or higher were included. Additionally, images with incorrect centration and motion or projection artifacts were excluded. 

### 2.3. Data Processing 

Acquired OCTA scans of the SCP were exported from the PlexElite Device in png format. Based on the detection method of Steger et al., ridge detection was used in Image J (National Institute of Health, Bethesda, MD, USA, Software Version 1.52p) to convert the images into a binary, skeletonized file [44]. 

The skeletonized image was then imported to MatLab (Release 2019b, MathWorks, Inc., Natick, MA, USA) to perform VTI calculation (Figure 1).

Therefore, VT was assessed with a mixed approach to measure vessel length and curvature. For each 6 × 6mm OCTA scan of the SCP, we selected all vessels for analysis. Therefore, all bifurcation and endpoints of the vessels were taken into processing. Vessel centerlines between each pair of bifurcation points were extracted through manual endpoint selection. The vessels’ endpoints were selected by simultaneous visualization of the grayscale and binary images. The grayscale image was used to identify bifurcations, and the binary image was used to locate the bifurcations more accurately with respect to the vessel boundaries and centerline. Afterward, the vessels could be segmented. Each vessel segmentation was defined manually, resulting in an individual VTI for each vessel segmentation in one OCTA image. After calculating the VTI of all vessel segments individually, the mean VTI was calculated per image using the VTI Calculation program created by Khansari et al. [42,45].

### 2.4. Statistical Analysis

Descriptive statistics are presented as counts and frequencies for categorical data and mean, including standard deviation, for metric variables. 

Results are presented as differences in mean VTI with corresponding 95% confidence intervals and *p*-values. *p*-values and confidence intervals are adjusted using Tukey’s method.

A *p*-value < 0.05 is considered significant. Overall *p*-values correspond to the T-test for means and Chi-squared or exact Fisher’s test. To compare mean VTI between study groups, an ordinary linear regression model was used. Additionally, age was included as a covariate, hence results are adjusted for these variables. All evaluations were carried out using the statistical software R (Version 4.0.4, R Foundation for Statistical Computing, Vienna, Austria).

## 3. Results

In total, we imaged 62 eyes of 62 participants (34 in the FD cohort and 28 in the HC cohort). Two eyes of the FD cohort and four eyes of the HC cohort had to be excluded due to the low quality of the images. Hence, 56 eyes of 56 patients, 32 in the FD cohort and 24 in the HC cohort, could be included for final analysis. The male:female ratio was overall 25:31 (12:20 in the FD cohort and 13:11 in the HC cohort). Overall, the mean age (±SD) was 37.7 (±13.56), 42.3 (±14.4) in the FD cohort, and 31.58 (±9.56) in the HC cohort. Overall, the mean VTI (±SD) was 0.21 (±0.07) in all assessed OCTA images, 0.23 (±0.09) in the FD cohort and 0.22 (±0.09) in the HC cohort, respectively. In the FD cohort, 18 patients had a classic type (male:female ratio 13:5), and 14 patients had a late onset of the disease (male:female ratio 7:7). An overview of the demographic data and the mean VTI (±SD) is provided in Table 1.

Comparison of mean and median VTI between FD cohort and HC cohort did not show significant differences.

Subgroup analysis showed that male patients with a classic type of FD had the highest mean, median, and individual data points VTI (0.33, SD ± 0.35), which was significantly higher compared to every other subgroup (*p*-value < 0.05, Table 2). No other comparison of VTI within the subgroup analysis showed significant differences (Table 2, Figure 2).

**Table 2 diagnostics-13-02496-t002:** Pairwise comparisons of vessel tortuosity index (VTI) between subgroups adjusted for age.

Comparison	Difference of Means	Lower/Upper Confidence Limit	*p*-Value
Classic (male)–Classic (female)	0.1203	0.01202/0.2286	0.02137
Classic (male)–Late-onset (male)	0.1414	0.01961/0.2633	0.01424
Classic (male)–Late-onset (female)	0.1464	0.0285/0.2644	0.007198
Classic (male)–Healthy (male)	0.1258	0.02012/0.2315	0.01115
Classic (male)–Healthy (female)	0.1118	0.002577/0.221	0.04201
Classic (female)–Late-onset (male)	0.02115	−0.0734/0.1157	0.9851
Classic (female)–Late-onset (female)	0.02616	−0.06917/0.1215	0.9636
Classic (female)–Healthy (male)	0.005543	−0.07676/0.08785	1
Classic (female)–Healthy (female)	−0.008498	−0.09877/0.08178	0.9998
Late-onset (male)–Late-onset (female)	0.005009	−0.1048/0.1148	1
Late-onset (male)–Healthy (male)	−0.0156	−0.1151/0.08389	0.9971
Late-onset (male)–Healthy (female)	−0.02965	−0.1369/0.07758	0.9624
Late-onset (female)–Healthy (male)	−0.02061	−0.1152/0.074	0.9868
Late-onset (female)–Healthy (female)	−0.03466	−0.1343/0.06501	0.9051
Healthy (male)–Healthy (female)	−0.01404	−0.09741/0.06933	0.996

Tukey’s adjustment was applied for all comparisons. Results are presented as differences of means with 95% CI and *p*-values.

**Figure 2 diagnostics-13-02496-f002:**
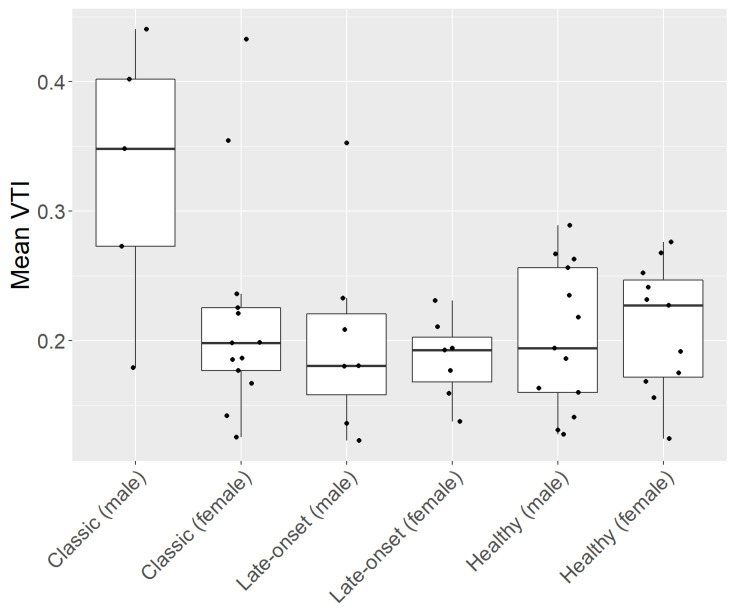
Subgroup analysis of patients with Fabry disease (FD) and healthy controls (HC). FD cohort consisting of patients with a classic and a late onset of the disease. Each subgroup is divided by gender (male and female).

## 4. Discussion

In this study, we analyzed retinal VT with a quantifying method by calculating the mean VTI of all vessels in the SCP per OCTA image of a group of patients with FD (FD cohort) compared to a healthy control group (HC cohort).

Regarding vessel selection, we selected all the vessels of the SCP per OCTA image. Vessel centerlines between each pair of bifurcation points were extracted through manual endpoint selection. To analyze the Vessels and calculate the VTI, we used the method introduced by Khansari et al. [42]. VTI calculation was computed for each centerline. 

Our results showed that male patients with classic FD had the highest mean, median, and individual data points of VTI compared to all other subgroups. As male patients with classic FD usually have a more severe progression of the disease compared to other subgroups of patients with FD, retinal vasculature can be more altered in these patients.

VT in patients with FD is speculated to be a result of intracytoplasmic storage of Gb3 within the endothelial cells, which can lead to endothelial proliferation and weakening of the vessel walls [1,13,32]. Hence, this could lead to a higher retinal VTI in these patients [46]. The finding of our study confirms data from previous studies where cohorts of patients with FD were analyzed to assess retinal VT with different methods [1,46].

Studies have shown that further changes in retinal vasculature and structure were found in patients with FD, e.g., an enlarged foveal avascular zone (FAZ) or increased vessel density (VD) and vessel length density (VLD) in the SCP and deep capillary plexus (DCP) in patients with FD compared to HC, but also hyperreflective foci in OCT images of patients with FD as a sign of accumulation of Gb3 in the retina [13,14]. In a previous study, we have already shown an association between Gb3 and OCTA VD and VLD [16]. This supports the hypothesis that quantitative OCTA parameters might be useful as diagnostic biomarkers for evaluating systemic involvement in FD, and possibly other diseases. However, we could only detect associations of Gb3 and VD as well as VLD changes in the SCP in the male subgroup of patients with FD. 

As stated in previous studies, Gb3 accumulation plays an essential role in the pathogenesis of FD and the severity of clinical manifestation in patients with FD can vary enormously depending on the activity of GAL and thereby the accumulation of Gb3 in several tissues’ lysosomes and fluids [1,5,6,7].

Retinal VT has been described in many systemic diseases and genetic disorders, such as systemic hypertension, diabetes mellitus, chronic anemia, and retinopathy of prematurity [40]. Regarding VTIs, there have been several approaches described to calculate a VTI, yet there has been no consensus on a standardized method [47,48,49,50,51,52,53,54,55]. VTI is a unitless index. The minimum value for VTI is equal to zero for an ideal straight line. Thus, the value of the VTI increases with more curvatures. In theory, there is no maximum value for VTI since it increases with the number of angles [40].

There are several approaches to calculating a VTI, whereas the main approaches can be divided into a distance approach, a curvature approach, or a mixed approach. The distance approach calculates the tortuosity as a quotient of the length of a vessel (arc length) over a straight line (chord length) between the two endpoints or branching points of the measured vessel or vessel section respectively.

For the curvature approach, an index is calculated that provides the minimum of changes in direction of the blood flow within a tortuous vessel. This approach has also several alternative ways of being calculated, as described by Abdalla et al. [39]. A remarkable variation is an approach that uses the sum of angles in their calculations. This method was initially proposed by Smedby et al. and, e.g., used by Sodi et al. [6,56]. It measures VT by evaluating the angles between a point on the curve and two other intersection points, with a predetermined vessel distance. If the vessel is close to a straight line, the sum of angles will be low, the result is normalized over the length of the vessel. 

The mixed approaches are methods that combine both previously described approaches, such as the one proposed by Khansari et al. [42]. This approach is sensitive to small changes in retinal VT, which makes it suitable for VT analysis in OCTA. Due to these reasons, we decided to use this approach in our study, taking both the curvature approach and the distance approach into account and being suitable for the analysis of OCTA imaging. [42]. We analyzed all retinal vessels per 6 × 6mm OCTA image of the SCP in order to avoid an unwilling selection bias of retinal vessels.

While being a large cohort of patients for a rare disease such as FD that received OCTA imaging, the number of cases remains low, especially in our subgroup analysis. In addition, we had to work with a non-age-matched cohort for comparison, limiting the comparability between the two cohorts. A further limitation is this study’s retrospective nature and the initially stated situation of various approaches in the literature regarding the assessment of retinal VT, making it difficult to compare our findings to other examples from the literature. Yet, we decided to take the advanced approach of a mixed model, as outlined above, to calculate the mean VTI per 6 × 6 mm OCTA image of all vessels in the SCP. However, we only assessed structural changes of retinal vessels with this approach, whereas no statement can be made regarding the dynamics of the retinal blood flow or further structural changes in the retinas of patients with FD. Nevertheless, OCTA has proven a reliable noninvasive imaging modality for determining microvascular changes in the retina, being a modality for screening and follow-up examinations in patients with retinal vascular diseases, including patients with common systemic diseases, but also patients with rare systemic diseases such as FD [57,58,59,60]. Only a few studies have performed an analysis of VTI in patients with FD, supporting the possibility of OCTA imaging as a valid biomarker for predicting systemic involvement in FD in the future [1,6,46]. 

## 5. Conclusions

We could detect a significantly higher retinal VT in male patients with classic FD compared to all other subgroups in our study. Further investigations of retinal VTI in patients with FD could strengthen the role of OCTA as a noninvasive screening and follow-up imaging modality to assess disease severity and progression in affected patients.

## Figures and Tables

**Figure 1 diagnostics-13-02496-f001:**
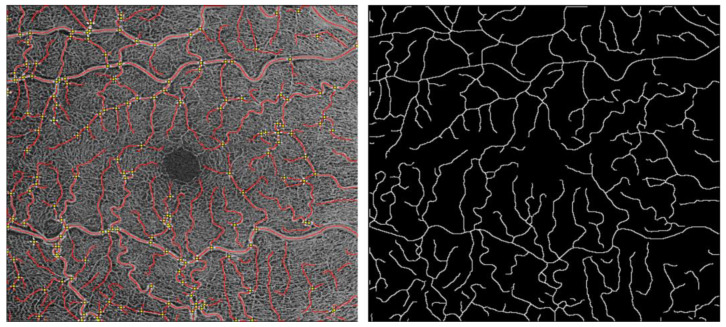
Process of measuring VTI: image binarization of the OCTA images acquired using Image J (National Institute of Health, Bethesda, MD, USA, Software Version 1.52p).

**Table 1 diagnostics-13-02496-t001:** Age and mean VTI of the Fabry disease (FD) cohort and healthy cohort.

	Classic Males	Classic Females	Late-Onset Males	Late-Onset Females	Healthy Males	Healthy Females	All Eyes	*p*-Value
Number (*n* = 56)	5	13	7	7	13	11	56	<0.001
Mean age in years (SD)	34.4 (15.6)	44.5 (11.8)	48.1 (5.64)	38.1 (21.8)	34.2 (9.64)	28.5 (8.88)	37.7 (13.6)	0.011
Mean VTI (SD)	0.33 (0.10)	0.22 (0.08)	0.20 (0.08)	0.19 (0.03)	0.20 (0.06)	0.21 (0.05)	0.22 (0.07)	0.015

Abbreviations: SD—standard deviation; VTI—vessel tortuosity index.

## Data Availability

Data will be made available upon request to the corresponding author.

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
