# Peer review of "Assessment of Retinal Vessel Tortuosity Index in Patients with Fabry Disease Using Optical Coherence Tomography Angiography (OCTA)"

_diagnostics, 2023, doi:10.3390/diagnostics13152496_

Round 1
Reviewer 1 Report (Previous Reviewer 2)
The authors applied a vessel tortuosity index to FD patients and healthy subjects and evaluated their difference in OCTA images. When I saw the statement in the abstract "...we assessed a novel retinal VT index (VTI)..." I thought the work might be related to proposing a new tortuosity index. After I got into more details I found the VTI used in the study is from 2017. Therefore, the adjective 'novel' may not be appropriate to describe the work or more judgment may need to claim the relativity of novelty.
In line 57 it says "It has been suggested that VT can be of diagnostic and prognostic value in patients with FD [5,11-16]", and the work also studied the VT of FD patients versus healthy subjects. The value of the work should be emphasized from a more holistic aspect when mentioning similar previous works.
In line 92-94 the authors mentioned there are several approaches proposed to quantify VT. What is the reason to choose [42]? If the main finding is that male FD patients have higher VT than all other subgroups by using [42], does it still hold when using other VTIs? It will be nice to use another VTI to support the finding.
Line 153: Unexpected line break
In line 58 and 63, the format of the citation is xxx [xx], but in line 57 and 59, the format becomes xxx[xx]. Whether to plug in a space or not is inconsistent throughout the whole manuscript.
Line 250: "Abdalla et al[39]" --> Abdalla et al.[39]
Author Response
Dear Reviewer
Thank you for your valuable comments and suggestions. Please find hereby our point by point reply.
The authors applied a vessel tortuosity index to FD patients and healthy subjects and evaluated their difference in OCTA images. When I saw the statement in the abstract "...we assessed a novel retinal VT index (VTI)..." I thought the work might be related to proposing a new tortuosity index. After I got into more details I found the VTI used in the study is from 2017. Therefore, the adjective 'novel' may not be appropriate to describe the work or more judgment may need to claim the relativity of novelty.
Thank you for pointing this issue out. We adapted the manuscript accordingly (Line 102).
In line 57 it says "It has been suggested that VT can be of diagnostic and prognostic value in patients with FD [5,11-16]", and the work also studied the VT of FD patients versus healthy subjects. The value of the work should be emphasized from a more holistic aspect when mentioning similar previous works.
Thank you for your suggestion. We tried to imply your suggestion and adapted the manuscript accordingly by adding a more holistic purprose off assessing VT (Line 74-76).
In line 92-94 the authors mentioned there are several approaches proposed to quantify VT. What is the reason to choose [42]? If the main finding is that male FD patients have higher VT than all other subgroups by using [42], does it still hold when using other VTIs? It will be nice to use another VTI to support the finding.
Thank you for your comment. We elaborated this issue now under the discussion section, explaining more detailed why we decided to use the method of Khansari et al. instead of another approach of other VTI approaches. Mainly, the approach of Khansari et al. is compared to other current VTI approaches very sensitive to small changes in retinal VT, which makes it suitable for VT analysis in OCTA. Moreover, it is a mixed model approach taking the calculations of both the curvature approach and the distance approach into account, enabling an overall higher accuracy in retinal VT assessment then other approaches.
Line 153: Unexpected line break
Thank you, we adapted the line break accordingly.
In line 58 and 63, the format of the citation is xxx [xx], but in line 57 and 59, the format becomes xxx[xx]. Whether to plug in a space or not is inconsistent throughout the whole manuscript.
Thank you, we adapted the citation format accordingly.
Line 250: "Abdalla et al[39]" --> Abdalla et al.[39]
Thank you, we corrected this accordingly.
Reviewer 2 Report (New Reviewer)
Dear authors,
congratulations on the interesting research and on the objective consideration of the limitations of the research that could not be avoided. My only complaint could possibly be the manual selection of bifurcation and endpoints. I hope that all bifurcation and andpoints on all OCTA images are taken into processing.
Author Response
Dear Reviewer
Thank you very much for your constructive feedback. Regarding the bifurcation and endpoints, all bifurcation and endpoints in each OCTA image were taken into processing. Vessel selection was performed analog to the method introduced by Khansari et al. We selected all the vessels of each image. Vessel centerlines between each pair of bifurcation points were extracted through manual endpoint selection. The vessel endpoints were selected by simultaneous visualization of the grayscale and binary images. On the grayscale image, we identified bifurcations and the binary image was used to locate them more accurately with respect to the vessel boundaries and centerline. Calculating all vessels per image, we could provide more reliable results compared to analyzing just some selected vessels per image. We added this information in the manuscript accordingly unter the section 2. Material and Methods, subsection 2.3 Data processing.
Round 2
Reviewer 1 Report (Previous Reviewer 2)
The current revision is satisfactory.
This manuscript is a resubmission of an earlier submission. The following is a list of the peer review reports and author responses from that submission.
Round 1
Reviewer 1 Report
Abstract:
1. „compared to all other subgroups” – please expand to be more clear for the reader
2. Please provide a one sentence rationale for the final statement:
“Further investigations of retinal VTI in patients with FD could be helpful 35 to use OCTA as a non-invasive screening and follow-up modality to assess disease progression in 36 affected patients.”
Manuscript
1. Please , somewhere at the beginning of the Introduction, provide info on the typical age of onset /diagnosis of FD. Info about childhood onset appears too late in the manuscript and is not really clear.
2. “The deficient or absent activity of GLA causes an accumulation of Globotrioasylceramide 50 .....etc” – please expand information on vascular consequences of such process, especially in the context of parameters measured by the OCTA. Authors concentrated on superficial capillary plexus in their OCTA measurements (as this was the software limitation I think). Please refer pathophysiology in FD to this particular plexus.
3. Please provide the formula for the VTI, not only description. OCTA naive reader should understand the idea of that parameter. By the way shouldn’t it be VTI from the start not VT ?
4. Control group: authors state : Non-age matched healthy subjects. Does that mean that the control group was not matche for age with the study cohort ? Why ? This an important drawback of the study.
5. Vessel distance ? Do authors mean vessel length ?
6. Statistical analysis, material and methods: please provide rationale behind separate analysis for males and females.
7. Lines 231-240. I think that it would be better to move this paragraph to introduction – when authors discuss the formula of VTI.
8. Please mention the lack of age adjustment between study and control groups in the limitation section. Please comment if it is significant.
9. English proofreading would help.
Reviewer 2 Report
L156-157: "the mean VTI was calculated per image using the VTI Calculation program created by Khansari et al.[42,45]" -> Only [42] is related to Khansari's work. [45] is even earlier than [42].
L186-187: This finding is against the results in Fig.2 of [46]. However, the discussion or investigation of the inconsistency seems to be missing.
L188: "Subgroup-analysis showed that male patients with a classic type of FD had the highest mean, median, and individual data points VTI (0.33, SD ± 0.35)" -> It is the female group that has the mean VTI 0.33. The following table 2, Fig. 1, and their relevant descriptions need to be checked for consistency.
L216-217: If the other two papers [1,46] (published in 2017 and 2021) already examined the relationship of patients with FD and their retinal VT, then the manuscript only uses another index ([42] published in 2017) to redo this work. In L25 it says, "In this study, we assessed a novel retinal VT index (VTI)...", the novelty of using the VTI here is questionable. A detailed comparison of this VTI with [1,46] or others should be done to demonstrate the novelty.
L298: Wrong reference format
L395: "(accessed on" -> It seems the sentence is not ended properly.